# Development and Validation of Prenatal Physical Activity Intervention Strategy for Women in Buffalo City Municipality, South Africa

**DOI:** 10.3390/healthcare9111445

**Published:** 2021-10-26

**Authors:** Uchenna Benedine Okafor, Daniel Ter Goon

**Affiliations:** 1Department of Nursing Science, University of Fort Hare, East London 5201, South Africa; 2Department of Public Health, University of Fort Hare, East London 5201, South Africa; dgoon@ufh.ac.za

**Keywords:** physical activity, strategies, pregnant women, SWOT analysis, BOEM plan, PESTEL analysis

## Abstract

Women rarely participate in physical activity during pregnancy, despite scientific evidence emphasising its importance. This study sought to develop an intervention strategy to promote prenatal physical activity in Buffalo City Municipality, Eastern Cape Province, South Africa. A multi-stage approach was utilised. The Strength, Weakness, Opportunity and Threat (SWOT) approach was applied to the interfaced empirical findings on prenatal physical activity in the setting. Subsequently, the Build, Overcome, Explore and Minimise model was then used to develop strategies based on the SWOT findings. A checklist was administered to key stakeholders to validate the developed strategies. Key strategies to promote prenatal physical activity include the application of the Mom-Connect (a technological device already in use in South Africa to promote maternal health-related information for pregnant women) in collaboration with cellphone and network companies; the South African government to integrate prenatal physical activity and exercise training in the medical and health curricula to empower the healthcare providers with relevant knowledge and skills to support pregnant women in prenatal physical activity counselling; provision of increased workforce and the infrastructure necessary in antenatal sessions and antenatal physical exercise classes and counselling; the government, in partnership with various stakeholders, to provide periodical prenatal physical activity campaigns based in local, community town halls and clinics to address the lack of awareness, misrepresentations and concerns regarding the safety and benefits of physical activity during pregnancy. The effective implementation of this developed prenatal physical activity by policymakers and health professionals may help in the promotion of physical activity practices in the context of women in the setting.

## 1. Introduction

Contrary to previously held negative beliefs, prenatal physical activity is associated with numerous health benefits, including physical, psychological and social advantages of physical activity during pregnancy, which are widely reported in the literature [1,2,3,4,5,6,7]. Nonetheless, while scientific evidence emphasises these benefits, most pregnant women do not participate. Furthermore, there is a considerable decline worldwide in physical activity among this special population [8,9,10,11,12,13]. In contrast to this trend, health professionals are advising women to partake in regular physical activity both during and after pregnancy, unless there are relevant medical or obstetrical complications [14]. Consequently, researchers have conducted various studies across countries and regions in an attempt to understand why pregnant women reduce or never take part in physical activity [8,9,10,11,12,15,16,17,18,19,20,21,22]. Unfortunately, most of these studies conducted are in developed countries, and there is very little information available within an African context [23]. Moreover, within the specific context of South Africa, the literature does not contain any research into the development of prenatal strategies that promote physical activity practices in pregnant women. In order to address this lack, our empirical findings on different aspects of prenatal physical activity in pregnant women, and midwives’ views of related counselling [24,25,26,27] informed and supported the need to develop and validate intervention strategies to promote physical activity and exercise during pregnancy in Buffalo City Municipality, Eastern Cape Province, South Africa.

Presumably, pregnancy serves as an opportune period to encourage a healthy lifestyle in individuals, since pregnant women are seemingly receptive to health messages [28]; therefore, a strategy to promote physical activity in pregnancy may prove useful to health professionals responsible for the antenatal healthcare of women. Strategy, as a term, refers to ‘what one may use to reach goals’ [29]. Strategically, a SWOT and/or Political, Economic Growth, Socio-Cultural, Technological, Laws and Environmental (PESTLE) analysis investigates both internal and external factors influencing pregnant women’s participation in physical activity. A SWOT analysis assesses the Strengths (S) and Weaknesses (W) as internal capabilities of an organisation/institution in contrast to Opportunities (O) and Threats (T) caused by the external environment [29,30,31]. This type of strategy entails the careful weighing of internal factors that enable the successful achievement of aims, which are termed ‘strengths’ or ‘weaknesses’ [29,30,31]. On the other hand, external factors are variables outside the reach of control, which constitute ‘threats’ or ‘opportunities’ related to the set goal [29,30,31]. In this particular SWOT analysis, since health providers assume the role of the strategic planner, ‘strengths’ or ‘weaknesses’ are identified as those factors relating to the provision of prenatal physical activity advice and health providers’ capability to deliver such advice. Subsequently, ‘opportunities’ or ‘threats’ are those factors of prenatal physical activity that mostly relate to pregnant women, such as behaviour, motivation or ability to engage in prenatal physical activity. In addition, the PESTLE model’s analytical tool is useful in identifying and analysing the fundamental drivers of change that operate in the external environment, which is useful in research reports [32]. The Build, Overcome, Explore and Minimize (BOEM) strategic model leverages on building strengths, overcoming weaknesses, exploring opportunities and minimizing threats, and in this case, factoring in the components of the BOEM to enhance prenatal physical uptake. The BOEM framework indicates the need to develop strategies that overcome mistrust, financial and human resources deficit, barriers to access and utilise physical activity facilities by pregnant women, while also eliminating stereotypical myths and beliefs, and minimising misuse of resources and ensure collaborative efforts by various stakeholders in promoting prenatal physical activity. Therefore, these strategic tools were used to comprehensively explore the various factors that define the facilitators of, or barriers to prenatal physical activity. Gaining insight into these environments would help minimise threats to prenatal physical activity while, at the same time, maximising opportunities for prenatal physical activity participation. In this article, we present the development and validation of the physical activity intervention strategy to encourage and promote prenatal physical activity and exercise within the context of the Eastern Cape by applying the SWOT, PESTLE and BOEM strategy models as part of the development and validation of a physical activity intervention strategy.

This study will, potentially, add to the body of knowledge on maternal health and further guide physical activity interventions that seek to address the needs of pregnant women in South Africa.

## 2. Methods

A multi-stage approach was utilized, with Phase I focussing on the empirical findings on facilitators of or barriers to prenatal physical participation in Buffalo City Municipality, Eastern Cape Province, which then lays the groundwork for the development of a prenatal physical activity strategy [24,25,26,27]. Therefore, the results of the studies above have provided us with guidance into developing an effective prenatal physical activity strategy that promotes the practice of prenatal physical activity. A study protocol was previously published on the development and validation of physical activity intervention strategy for pregnant women in Buffalo City Municipality, South Africa [33].

The SWOT, PESTLE and BOEM models underpin the development of the proposed intervention for the promotion of prenatal physical activity in Buffalo City Municipality, Eastern Cape Province. SWOT analysis is applicable to many health-related issues, including strategies for future healthcare [34]. Figure 1 presents the SWOT, PESTLE and BOEM strategies used to develop prenatal physical activity strategies.

As previously pointed out, the SWOT analysis strategy assesses a situation to determine the ‘strengths’, ‘weaknesses’ that internally characterise a system, which needs to be addressed, respectively, while ‘opportunities’ and ‘threats’ are viewed as external factors that may be used or avoided [35]. For the sake of this study, the SWOT tool was used to evaluate and manage the internal and external factors affecting the uptake of prenatal physical activity and the provision of prenatal physical services at primary healthcare clinics in Buffalo City Municipality. Similarly, the PESTLE analytical tool analyses a particular situation to identify actions that may cause failure. In addition, the researchers utilised the BOEM approach to develop a prenatal physical activity strategy by building on the identified strengths, overcoming the weakness, exploring the opportunities and minimising the threats to physical activity [36]. In order to identify the relevant strengths and weaknesses of prenatal physical activity participation in the setting, the researchers conducted various interviews with experts, and consulted published and/or unpublished data from the Department of Health. An external assessment (or analysis) was used to identify the opportunities and threats [35]. Table 1 presents the action plan indicating the BOEM developing strategies based on SWOT analysis.

### 2.1. Validation of the Developed Intervention Strategy

Phase I and II of the study preceded the validation of the developed intervention strategy. Validation seeks to ascertain the empirical credibility of a scientific model of a particular discipline [37], and there are several ways of validating empirical knowledge. One alternative approach is to note and share a thought or opinion pertaining to what a phenomenon is and operates, without conducting formal research to test the views [37]. We analysed the empirical findings synergistically with SWOT and PESTLE analytical strategic frameworks in order to identify pointers that would aid the validation of the developed strategy deemed feasible, effective and sustainable to address the gaps identified during the research.

A purposive sample of seven professional academic experts with extensive knowledge and proven academic and scholarly background on prenatal physical activity and maternal health were selected to participate in the validation process. They include gynaecologists, obstetricians, midwives/professional nurses and exercise physiologists. The findings from Phase I, the SWOT and PESTLE analyses, as well as the BOEM model and the subsequently developed strategies were presented to the professional academic experts. They were requested to provide critical comments on the developed strategy in the context of needs of the pregnant women, determine the feasibility of the strategy in promoting prenatal activity in women in the locality. Therefore, the feedback from the experts help to modify the strategy, which was then presented to the key stakeholders for validation.

### 2.2. Key Stakeholder Consultation

Key stakeholders that were consulted were mainly maternity health team managers, such as physician, obstetrician, midwife, paediatric nurse and primary healthcare nurse. In each of the 12 selected antenatal health clinics, two midwives and pregnant women were purposively chosen to participate in the validation process. In all, 54 participants participated. The managers of these particular healthcare facilities were also included since they are the agents who would implement or supervise the application of these developed strategies in their respective healthcare facilities. Moreover, it is important to also include pregnant women since patient engagement may provide insights into possible contextual interventions strategies that are relevant to addressing the individual needs of pregnant women in order to promote their physical activity. Furthermore, a checklist of the developed strategies was utilised to solicit the opinion of and information from the above-mentioned stakeholders.

The empirical findings of the study (Phase I) and the developed strategies based on the SWOT/PESTLE analysis were presented to the relevant participants to discuss, deliberate on and provide their comments and opinions regarding its feasibility, accessibility and sustainability. The stakeholders’ views were analysed and then used to perfect the accepted strategy for the promotion and implementation of prenatal physical activity within the context of Buffalo City Municipality the Eastern Cape.

### 2.3. Data Analysis

A checklist was created to determine the possible strategies to facilitate physical activity promotion based on the PESTLE/SWOT analytical matrix; the comments and opinions on the checklist from the experts and stakeholders were analysed using frequency and percentage counts, and where applicable, a content thematic analysis was applied. The unit of analysis was the different constructs in the PESTLE/SWOT framework. Firstly, the comments from the experts were analysed. Secondly, two strategies that proved irrelevant were removed, while three were merged. Then, the stakeholders’ comments and views were analysed and integrated into the developed strategy.

## 3. Results

There were very few comments and critique received in terms of the stakeholder’s analysis of the prenatal physical activity strategy to promote physical activity and exercise during pregnancy (Table 2); these comments and suggestions were incorporated into the strategy, accordingly. Table 3 presents stakeholders’ validation of the physical activity intervention strategies for the promotion of prenatal physical activity in Buffalo City Municipality. Of all the strategies endorsed by the health managers and midwives, the most highly endorsed strategy was the application of scientific and technological innovations to provide basic information of the benefits of prenatal physical activity to pregnant women with the use of the Mom-Connect device. This device is already used in South Africa and promotes maternal health-related information for pregnant women. Relatedly, stakeholders also highly recommended collaborative partnerships with the various cellphone and network companies operating in South Africa. These partners include companies such as Vodacom, MTN, Cell C and 8ta, which would assist in dissemination information to regarding the benefits of prenatal physical activity. In addition, the pregnant participants who were sampled in the validation process shared similar opinions. They were all excited about a partnership between the government and the networks to facilitate the communication of information on prenatal physical activity. Relatedly, the use of cellphone networks was supported since many people own and use cellphones.

These pregnant women vehemently supported the incorporation of antenatal physical activity and exercise classes into routine antenatal healthcare services. They indicated that ‘this will build in them, the spirit of team play and encourage us to embrace physical activity during pregnancy’. Furthermore, the midwives highly praised the feasibility of the interventional strategy, which includes the provision of periodic prenatal physical activity, exercise training and workshops for healthcare professionals in order to mitigate their lack of knowledge on prenatal physical activity recommendations and guidelines.

## 4. Discussion

Physical activity is one of the key elements in promoting maternal health during the prenatal and postpartum period; therefore, interventions strategies to drive this agenda are needed to encourage women to maintain an active lifestyle during and after pregnancy. However, prenatal physical activity and exercise have not been widely studied within the context of South Africa. To address this lack, a prenatal physical activity intervention strategy has been developed specifically for South Africa and is on the first empirical findings [24,25,26,27]. This timely move provides a unique contribution to, and lays the groundwork for, future prenatal physical activity intervention strategies, not only in the Eastern Cape Province but throughout South Africa as a whole.

The most notable intervention strategy to promote physical activity in pregnant women in this setting is the application of scientific and technological innovations to provide basic information on the benefits of prenatal physical activity to pregnant women by utilising the Mom-Connect device to provide maternal health-related information to pregnant women. As previously stated, the study’s stakeholders highly sanctioned a collaborative partnership with the various cellphone and network companies in South Africa, namely Vodacom, MTN, Cell C and 8ta, who assist in disseminating information regarding the benefits of prenatal physical activity. They further asserted that printing relevant information on airtime slips, and automatic voice messaging, provided when dialling or receiving calls, to create awareness on recommended physical activity during pregnancy. In order to reap the benefits of participation in prenatal physical activity entails the designing and implementation of corresponding interventions strategies that seek to address the needs of women during pregnancy. Accordingly, community health programmes tailored to suit pregnant women should include activities that would help in promoting awareness about physical activity during pregnancy [38]. However, the empirical findings showed that pregnant women in South Africa rarely, if ever, received physical activity counselling since prenatal healthcare providers themselves had little or no knowledge of prenatal physical activity, nor did they possess the skills by which to impart such information [25]. This lack on the part of healthcare providers is one of the main barriers to prenatal physical activity counselling [39,40,41]. Consequently, the government and other non-governmental bodies have a tremendous role to play in augmenting the efforts of the healthcare providers in this direction. In light of this, as sanctioned by the stakeholders in this present study, the existing Mom-Connect is deemed a good platform by which to incorporate prenatal physical activity messages encouraging and promoting physical activity among women in South Africa during and after pregnancy. This phone-based, technological device was developed by the South African National Department of Health to promote maternal and child health. Midwives emphasised the use of information technology, in this case, the Mom-Connect, to disseminate information about prenatal physical activity to pregnant women [27]. This is particularly concerning in this era of the COVID-19 pandemic. Even before the COVID-19 pandemic, the pregnancy stage presents unique challenges to being sufficiently active, and the COVID-19 pandemic has ushered in more difficulties in terms of limiting social contact and other opportunities for physical activity [42]. These prevailing circumstances suggest the need to explore the available technological applications and devices mentioned above to promote and influence women to be physically active during pregnancy. Notably, other studies have shown that including other strategies such as regular telephone reminders or meetings with the intervention deliverer, as well as encouraging regular physical activity at home, would help participants maintain healthy behaviours [43,44]. 

Another relevant strategy is that the government and policymakers integrate or infuse prenatal physical activity and exercise training into the medical and health curricula of existing higher institutions of learning concerned with the teaching of maternal health in the Eastern Cape Province since the research has found empirical proof that indicates prenatal healthcare providers lack the knowledge and skills, resulting in the inability to offer prenatal physical activity. Educational intervention is a key to change behaviour. A previous educational interventional study showed improvement in the professionals’ knowledge regarding leisure-time walking and women who were cared for by the intervention group were more likely to receive guidance regarding leisure-time walking [45]. A recent study combining education about physical activity with a training session on facilitating behaviour change emphasised the importance of educating health professionals on prenatal physical activity based on the findings from a needs assessment that demonstrates health professionals had limited knowledge about physical activity during pregnancy [46], which is likely attributed to limited availability of undergraduate and continued professional training of healthcare professions on prenatal physical activity [47]. Thus, incorporating prenatal physical activity into the medical and nursing training schools would empower the healthcare providers with the knowledge and requisite skills they need to assist pregnant women in terms of prenatal physical activity prescription and advice. Therefore, there is a ‘call to action’ for policy makers to integrate physical activity and exercise prescription as part of the curricula training of medical students [48,49], and across all health professions saddle with the responsibility of antenatal care [50], such as midwives [28,51] and exercise physiologists [52]. 

Another strategy that received overwhelming support from the stakeholders is that the government recruit more midwives to assist in antenatal sessions and antenatal physical exercise classes and counselling. However, the feasibility of this particular strategy hinges on addressing the current shortage of midwives by employing more of them to mitigate the challenge of busy schedules and many responsibilities of the antenatal health clinics. The dire shortage of this category of a healthcare provider is one of the reasons that prenatal physical activity is seldom a priority in primary health clinics in Buffalo City Municipality since midwives have to perform several other functions [27]. The limited human resources affect their ability to discuss physical activity with pregnant women and is compounded by the lack of antenatal exercise facilities/equipment for these women at the clinics. These institutional bottlenecks, directly or indirectly, have a negative impact on maternal health promotion, especially physical activity counselling [27]. Therefore, to promote the uptake of physical activity counselling associated with the shortage of health providers, there is a need to strengthen the primary healthcare clinics’ workforce and infrastructures.

Notwithstanding, another practical strategy endorsed by the stakeholders is the documentation of misconceptions and subsequent clarification of the safety concerns of prenatal physical activity. In addition, the document should be accessible to all women at the clinics in form of a small pamphlet or booklet. Various studies have found physical activity counselling through one-on-one sessions [44,53,54,55], and providing prenatal physical activity counselling and advice on the recommended physical exercise for pregnant women using information booklets, leaflets, and/or websites [53,55,56,57] are feasible interventions options that promote prenatal physical activity. Previous studies in South Africa have emphasised the utilisation of instructional resources, such as posters, brochures and Digital Video Discs (DVDs), as part of an intervention strategy to promote physical activity in antenatal clinics [58,59]. In a recent study, pregnant women advocated the inclusion of exercise brochures and videos [60,61], which tend to suggest that pregnant women are aware of what they require to be physically active; therefore, context-specific interventions to accommodate their needs or concerns on physical activity during pregnancy are essential. The ultimate goal of any strategy is to effect a change in behaviour and attitude to prenatal physical activity; therefore, incorporating behaviour change techniques into interventions may be helpful in improving physical activity levels during pregnancy [62]. In this regard, providing prenatal physical education and counselling would help educate women concerning the importance of prenatal physical activity practice, and is seemingly most effective through one-on-one approach [62]. A recent systematic review has identified individual interviews, group interviews, access to information through brochures or multimedia supports and use of smartphone applications for personal training and general information as interventions to promote physical activity during pregnancy [60]. The review further indicated that individual interventions are commonly used, and are reinforced by reminders during routine consultations, or through emails or informative brochures [60]. It has been shown that an individualised intervention incorporating two telephonic reminders increased physical activity participation in overweight and obese pregnant women [63]. 

In addition, we reason that if periodic prenatal physical activity campaigns are presented in local community town halls and clinics by the government in synergy with the other stakeholders such as NGO’s and faith-based organisations, this would aid and address the lack of awareness, misrepresentations and concerns around the safety and benefits of physical activity during pregnancy. Koleliat et al. [61] support designing a community-based intervention in which pregnant women can interact and share their challenges and concerns.

### 4.1. Implications

We envisaged that the barriers to prenatal physical activity among women in this particular setting would be addressed through this present, developed and validated physical activity strategy to facilitate and promote the practice of prenatal physical activity for better maternal health outcomes. This strategy captures several different prenatal physical activity perspectives within the context of Buffalo City Municipality in the Eastern Cape Province, South Africa. If implemented, it would aid context-specific interventions that could address the needs of pregnant women; furthermore, it would assist in motivating them, to engage in physical activity during pregnancy, and would support them in this endeavour. Therefore, the implementation of the developed prenatal physical activity strategy could improve physical activity practice of women.

### 4.2. Strengths

This context-specific prenatal physical activity strategy would potentially provide insight and direction on how to address the impediments to physical activity during pregnancy identified in this studied region, and how to promote physical activity in pregnant women. In addition, the developed physical activity strategy was based on a prospective evaluation of physical activity in an under-researched, poor and resource-constrained setting; therefore, it sets the pace for future studies investigating on physical activity interventions strategies aimed at promoting prenatal physical activity in this particular geographical area of South Africa, and even outside of it. 

## 5. Conclusions

The application of scientific and technological innovations to provide basic information on prenatal physical activity by utilising the Mom-Connect, a technological device already available in South Africa, to promote maternal health-related information for pregnant women is a feasible, effective and sustainable strategy to promote prenatal physical activity within the BCM setting. This strategy can work effectively in synergy and collaboration with the various cellphone and network companies operating in South Africa, such as Vodacom, MTN, Cell C and 8ta, in order to assist the spread of information related to the benefits of prenatal physical activity. In addition, offering professional development courses for health professionals involved in maternity care in the Eastern Cape Province would empower healthcare providers with the necessary basic knowledge and requisite skills required to assist pregnant women with effective prenatal physical activity advice or counselling. The recruitment of more midwives (by the government) to assist in antenatal sessions and antenatal physical exercise classes and counselling is another strategy that received overwhelming support from the stakeholders. However, to address the shortage of health providers, as its impact on prenatal physical activity counselling, there is a need to strengthen primary healthcare clinics’ workforce and infrastructures. In addition, if the government, in collaboration with other stakeholders, provides periodic prenatal physical activity campaigns, the lack of awareness, misrepresentations and concerns regarding the safety and benefits of physical activity during pregnancy could be effectively addressed. The above-mentioned prenatal physical activity strategies provide direction for policymakers and health professionals in the promotion of prenatal physical activity practices of women and, if implemented, may further improve their maternal health outcomes within the Buffalo City Municipality setting.

## Figures and Tables

**Figure 1 healthcare-09-01445-f001:**
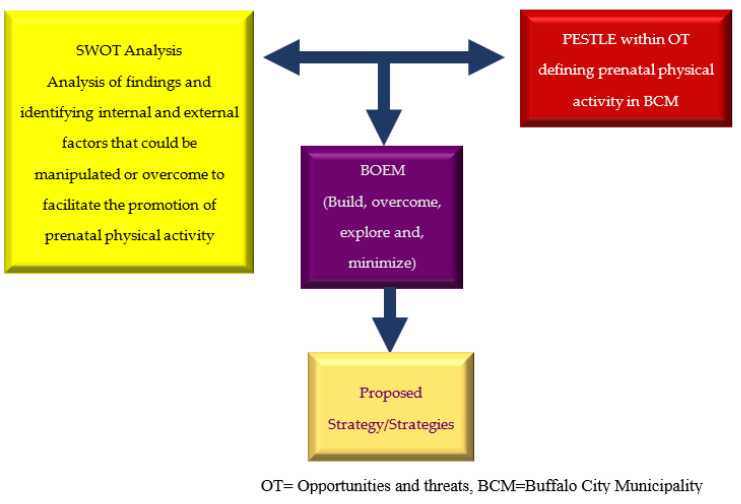
SWOT, PESTLE and BOEM strategies.

**Table 1 healthcare-09-01445-t001:** The BOEM strategy from the SWOT analysis of prenatal physical activity among women in the Eastern Cape.

	Strengths	Weaknesses
INTERNAL	Effective allocation of antenatal resources Availability of medical and health sciences departments to offer prenatal physical activity courses locally Free antenatal healthcare provision All available health centres provide antenatal healthcare services Government and research bodies, such as the National Research Foundation of South Africa and the South Africa Medical Research Council, support of research and development activities related to maternal health	Insufficient number of healthcare providers (midwives) to provide antenatal healthcare services, including prenatal physical activity advice or exercise classes Training on and knowledge about prenatal physical activity is sorely lacking Financial support for prenatal physical activity is lacking Prenatal physical activity advice, counselling or information is often unavailable Prenatal physical activity class or sessions are not provided for Patients are not referred for prenatal physical activity Prenatal physical activity is not a prioritised component of the antenatal primary healthcare service Prenatal physical activity guidelines do not exist There is no training course for midwives on prenatal physical activity No individualised prenatal physical activity counselling to pregnant women
EXTERNAL	Threats Inadequate material to provide prenatal physical activity services No political funding of prenatal physical training for midwives and organisation of prenatal physical activity community campaigns Human and material resources are scarce, which constrains the provision of prenatal physical activity services Unemployment in women result in the following: ○ There is no money to pay for gym exercise ○ There is no money to pay transport costs to and from the gym ○ There is no money to hire or employ a house helper ○ There are changes in household responsibilities Poor support (family, friends and peers) of pregnant women in terms of prenatal physical activity Women rarely participate in physical activity during pregnancy Women hold negative beliefs about the safety of physical activity and exercise during pregnancy Women are uneducated Women are underprivileged Women have a high unemployment rate The main source of women’s income is social grants Social media is absent and cannot be used to promote prenatal physical activity Technological resources for communication are lacking Weather conditions are unsafe or unfriendly Women have misconceptions about prenatal physical activity Neighbourhoods are unsafe for outdoor physical activity Recreational and outdoor physical activity is lacking COVID-19 pandemic challenges	Opportunities Government has a desire to provide primary health care services more accessible to people Mom-Connect technology is available to provide maternal health information Other stakeholders (non-governmental organisations (NGOs), faith-based organisations, community) are partnering with government concerning women’s maternal health
	Legal There is a lack of policy and guidelines on prenatal physical activity for possible interventions	Legal Strengthen maternal basic health education Initiate prenatal physical activity or exercise training courses and workshops to address the lack of knowledge of healthcare professionals, particularly among midwives who have limited knowledge and information on prenatal physical activity and exercise

**Table 2 healthcare-09-01445-t002:** Experts analyses and comments on the developed prenatal physical activity strategy.

Developed Strategy Item	Comment/Critique/No Comment
Intervention strategy 1: Building on strengths	
Training of gynaecologists/obstetrics/midwives/nurses to provide prenatal physical activity during antenatal sessions, and other categories of health providers, such as physiotherapists, exercise physiologists, biokineticists (practice exclusively in South Africa and Nambia), responsible for women’s’ healthcare in order to offer prenatal physical activity counselling during antenatal or consultation sessions.	This is good but beyond their scope of practice. They can only provide physical activity counselling. Commendable. Necessary since physical exercise in pregnancy helps burn calories and assist in minimising excessive weight gain. This would be feasible and to be included in documentation.
Increased focus of health/medical education on the morbidity of excess weight gain and physical inactivity during pregnancy.	Agree. Too general. Good, but can be merge with prenatal physical counselling.
Provision of prenatal physical activity campaigns in the community, involving ward-based outreach teams, NGOs and faith-based organisations. To empower and motivate women, well-known, influential female public figures should be involved as guest speakers at outreach events to create awareness on physical activity participation during pregnancy.	Very practical. Feasible.
The inclusion of physical activity counselling by trained professionals and antenatal exercise classes as one of the components of antenatal healthcare.	Yes. I agree.
Intervention strategy 2: Overcoming of weaknesses	
Application of scientific and technological innovations to provide basic information about the benefits of prenatal physical activity. For example, the use of Mom-Connect, a technological device already in use in South Africa to promote maternal health-related information for pregnant women.	Yes. I agree. No comments. Very important. Mom-Connect was developed many years ago but is still not widely enough utilised. My suggestion would be to focus on one or two interventions, which seem to be more achievable since the platform is already in place and funded, though it would be necessary to conduct some monitoring and evaluation to determine whether this strategy is effective. I fully agree that the technological platform of Mom-Connect would be the best manner in which to communicate basic information.
Actively involve support groups (partners, family and friends) in prenatal care so that others who influence the mothers are empowered to ask questions, understand risks and have the same goals regarding the benefits of prenatal physical activity for maternal health of both the mother and the baby.	Yes. I agree. No comments. This is relevant and important. This is feasible.
Midwives to conduct exercise classes of suitable duration and intensity that are consistent with the recommended guidelines for pregnant women to maintain health.	Not feasible
Intervention strategy 3: Exploring opportunities	
Negotiate with corporate organisations or companies to include in their products slogans promotional information on prenatal physical activity. For example, refrigerator magnets, grocery plastic bags, sanitary pads, cosmetics, baby products, billboards along the road and public. Radio stations and television broadcasting should also be included.	Yes. I agree. Very practical and can be rolled out with ease. This is possible.
Collaborate with cellphone and network companies, such as Vodacom, MTN, Cell C and 8ta, and have them assist in the dissemination of information related to the benefits of prenatal physical activity. Furthermore, they can include such information on their airtime slips and set up automatic voice messages, either when dialling or receiving calls, to create awareness of the importance of recommended physical activity during pregnancy.	I agree. This is very helpful. This is an excellent strategy and all members of communities that can share the information use it.
Conduct continued research in the field of effective interventions that promote physical activity during pregnancy.	I agree. Very necessary. There must be scientific evidence of all interventions.
Provide healthcare providers with relevant physical activity education and counselling strategies that are culturally appropriate, sensitive and effective.	Yes. I agree. Correct.
Create and provide an opportunity for pregnant women and other support groups (family members, spouses/partners) to attend monthly group meetings that focus on physical activity. Make those meetings attractive by providing treats, snacks or games.	Helpful. Doable. Feasible, but it needs to be in close proximity to where antenatal care is received.
Intervention strategy 4: Mitigation of threats	
Document the misconceptions and clarifications about the safety concerns of prenatal physical activity. Make this document accessible to women at the clinics in the form of a small pamphlet or booklet.	Yes. I agree. No comments. This will provide a readily accessible information resource for the mothers. Feasible strategy. This will be good and make it accessible in formats that can be taken home such as a small pamphlet or booklet.
Implement individual face-to-face or group interviews emphasising the risks of a sedentary lifestyle and physical activity and provide information on the minimal physical activity required for pregnant women, as well as the importance of nutrition during and after pregnancy during routine consultations.	Difficult to implement. Not feasible considering lack of human resources.
Provide general health education training to home-based carers and community health workers in terms of the importance and benefits of prenatal physical activity, so that they are equipped to create more awareness within their respective communities about the need to participate in physical activity and exercise during pregnancy.	Yes. I agree. No comments. Very beneficial. This will be an excellent strategy to continue spreading awareness.
Periodic professional training of midwives, and keeping them updated on the latest prenatal physical activity and exercise prescriptions, contraindications, recommendations and guidelines; furthermore, as part of the nursing education training curriculum, policy-makers should further add to the maternal guidelines that address prenatal physical activity information and practice.	Yes. I agree. No comments. Training of healthcare providers is important.
The government should recruit more midwives at primary healthcare centres to assist in the antenatal sessions, this would include them providing antenatal physical exercise classes.	This will aid successful intervention if this is possible at the government level. This would be excellent.
Form a multi-disciplinary antenatal physical exercise team comprised exercise physiologists and biokineticists who can demonstrate antenatal exercises to women during their antenatal clinic visits.	Would be very helpful. This would be good, but they are not always involved in all health care facilities or available.
As a long-term strategy, build and procure antenatal physical exercise facilities and equipment to encourage and promote prenatal physical activity.	Would rather recommend physical activity that can be conducted with minimal equipment to encourage all women to continue exercising at home and not only during the antenatal clinic. Excellent intervention.

**Table 3 healthcare-09-01445-t003:** Developed physical activity intervention strategy.

Intervention Strategy 1: Building on Strengths	Responsibility
Provide knowledge to gynaecologists, obstetricians, midwives and nurses on prenatal physical activity counselling during antenatal sessions.	Eastern Cape Department of Health
Provide relevant prenatal campaigns within communities and involve ward-based outreach teams, NGOs and faith-based organisations. Utilise influential well-known female public figures as guest speakers at outreach events to create awareness of the benefits of physical activity participation during pregnancy and empower communities.	Eastern Cape Department of Health, in collaboration with the Primary Health Care Nurses/Midwives
Include physical activity counselling by trained professionals and antenatal exercise classes as one of the components of antenatal healthcare.	Primary Health Care Nurses/Midwives
Intervention strategy 2: Overcoming of weaknesses	
Utilise scientific and technological innovations to provide basic information on the benefits of prenatal physical activity. For example, the use of Mom-Connect, a technological device already available in South Africa, to promote maternal health-related information to pregnant women.	Eastern Cape Department of Health in collaboration with the Primary Health Care Nurses/Midwives
Actively involve support groups (partners, family and friends) in prenatal care so they are empowered to ask questions, understand risks and have the same goals regarding the benefits of prenatal physical activity for the maternal health of both the mother and the baby.	Primary Health Care Nurses/Midwives
Midwives should conduct exercise classes with duration and intensity consistent with the recommended guidelines for pregnant women to maintain health.	The Primary Health Care Nurses/Midwives
Intervention strategy 3: Exploring opportunities	
Negotiate with corporate organisations or companies to include in their products slogans promotional information on prenatal physical activity. For example, refrigerator magnets, grocery plastic bags, sanitary pads, cosmetics, baby products, billboards along the road and public. Additionally, radio stations and television broadcasting.	Eastern Cape Department of Health in collaboration with the Primary Health Care Nurses/Midwives
Collaborate with cellphone and network companies, for example, Vodacom, MTN, Cell C and 8ta, to assist in information dissemination regarding the benefits of prenatal physical activity. They can write information on prenatal physical activity on their airtime slips, provide automatic voice message either when dialling or receiving calls to create awareness on recommended physical activity during pregnancy.	Eastern Cape Department of Health in collaboration with the Primary Health Care Nurses/Midwives
Conduct continued and ongoing research into effective interventions that promote physical activity during pregnancy.	Research institutions, organisations and government-initiated and sponsored researches
Educate prenatal healthcare providers on physical activity education and counselling strategies that are culturally appropriate, sensitive and effective.	Primary Health Care Nurses/Midwives
Create and provide an opportunity for pregnant women and other support groups to attend monthly group meetings on physical activity.	Primary Health Care Nurses/Midwives
Intervention strategy 4: Mitigation of threats	
Documenting the misconceptions and clarifications about the safety concerns of prenatal physical activity and making the document accessible to women at the clinics in form of a small pamphlet or booklet.	The Primary Health Care Nurses/Midwives
Providing general health education training to the home-based carers and community health workers on the importance and benefits of prenatal physical activity, so that they would, in turn, create such awareness to the community about the need to participate in physical activity and exercise during pregnancy.	The Primary Health Care Nurses/Midwives
Periodic training and update of midwives on prenatal physical activity and exercise prescriptions, contraindications, recommendations and guidelines.	Eastern Cape Department of Health
Policymakers to add to the maternal guidelines prenatal physical activity information and practice as part of the nursing education training curriculum.	Eastern Cape Department of Health/Primary Healthcare antenatal managers
The government to recruit more midwives at the primary healthcare centres to assist in antenatal sessions, which will include providing antenatal physical exercise classes.	Eastern Cape Department of Health
As a long-term strategy, build and procure antenatal physical exercise facilities and equipment to encourage and promote prenatal physical activity.	Eastern Cape Department of Health

## Data Availability

Data available on request.

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
