# Peer review of "Development and Validation of Prenatal Physical Activity Intervention Strategy for Women in Buffalo City Municipality, South Africa"

_healthcare, 2021, doi:10.3390/healthcare9111445_

Round 1

Reviewer 1 Report

Thank you for giving me the opportunity to review this manuscript. This manuscript, entitled "Development and validation of prenatal physical activity intervention strategy for women in Buffalo City Municipality, 3 South Africa", aimed to develop and validate an intervention strategy to promote prenatal physical activity and exercise in women in City of Buffalo City, Eastern Cape Province of South Africa. The proposed study is interesting, however, I would like to draw the authors' attention to some methodological issues in an attempt to improve this manuscript.

Abstract: The Abstract must have a total of about 200 words at most according to Healthcare standards. Although the abstract must have a single paragraph, it must follow the style of structured abstracts, but without titles and containing the following items: Background: place the issue addressed in a broad context and highlight the objective of the study; Methods: Briefly describe the main methods or treatments applied. Include all relevant pre-registration numbers and species and strains of any animals used. Results: summarize the main findings of the article; and Conclusion.
So I advise a restructuring of the presented summary.

Introduction: Well contextualised.
Methods:
The methodology seems to me to be very confusing and still needs to be restructured.
Lines 94 to 99 cause confusion to what you intend to investigate. It is important that the first paragraph of the Methodology describes the type of research that will be carried out.
The objective presented at the end of the introduction indicates that physical activity intervention strategies will be developed and validated to encourage and promote prenatal physical activity, but in the first paragraph of the methodology it says that the development process
and the validation of prenatal physical activity has been published elsewhere. So I believe that either the objective or the methodology is not adequate.

The sentence: “The empirical findings from both the quantitative and qualitative approaches, the 94 reported earlier, formed the foundation for the development of a prenatal physical activity 95 strategy (Phase I)” seems to me very confusing and does not reflect the type of study that is presented.

The phrase “The process outling the development 98 and validation of the prenatal physical activity has been published elsewhere [33].” must be rewritten. My suggestion: “A Study Protocol was previously published on the development of physical activity intervention strategy for pregnant women in Buffalo City County, South Africa.

The study does not follow a standardized validation methodology. Kappa agreement analysis was not performed among the professionals evaluated. It should make clear which steps have already been taken and which have been carried out in this study.

Results and discussion
The results are unclear and the discussion does not reflect information from an instrument creation and validation study.

Author Response

Reviewer#1- Comments and Suggestions for Authors

Thank you for giving me the opportunity to review this manuscript. This manuscript, entitled "Development and validation of prenatal physical activity intervention strategy for women in Buffalo City Municipality, South Africa", aimed to develop and validate an intervention strategy to promote prenatal physical activity and exercise in women in City of Buffalo City, Eastern Cape Province of South Africa. The proposed study is interesting, however, I would like to draw the authors' attention to some methodological issues in an attempt to improve this manuscript.

Comment

Abstract: The Abstract must have a total of about 200 words at most according to Healthcare standards. Although the abstract must have a single paragraph, it must follow the style of structured abstracts, but without titles and containing the following items: Background: place the issue addressed in a broad context and highlight the objective of the study; Methods: Briefly describe the main methods or treatments applied. Include all relevant pre-registration numbers and species and strains of any animals used. Results: summarize the main findings of the article; and Conclusion.
So I advise a restructuring of the presented summary.

Response

The summary has been revised to 258 words.

Women rarely participate in physical activity during pregnancy, despite scientific evidence emphasising its importance. This study sought to develop intervention strategy to promote prenatal physical activity in Buffalo City Municipality, Eastern Cape Province, South Africa. A multi-stage approach was utilised. The Strength, Weakness, Opportunity and Threat (SWOT) approach was applied to the interfaced empirical findings on prenatal physical activity in the setting. Subsequently, the Build, Overcome, Explore and Minimise model was then used to develop strategies based on the SWOT findings. A checklist was administered to key stakeholders to validate the developed strategies. Key strategies to promote prenatal physical activity include: the application of the Mom-Connect (a technological device already in use in South Africa, to promote maternal health-related information for pregnant women) in collaboration with cellphone and network companies, the South African government to integrate prenatal physical activity and exercise training in the medical and health curricula to empower the healthcare providers with relevant knowledge and skills to support pregnant women in prenatal physical activity counselling, provision of increased workforce and the infrastructure necessary in antenatal sessions, and antenatal physical exercise classes and counselling, and the government, in partnership with various stakeholders, to provide periodical prenatal physical activity campaigns based in local, community town halls and clinics to address the lack of awareness, misrepresentations and concerns regarding the safety and benefits of physical activity during pregnancy. The effective implementation of this developed prenatal physical activity by policymakers and health professionals may help in the promotion of physical activity practices in the context of women in the setting.

Comment

Introduction: Well contextualised.

Response

Appreciated.

Comments

Methods: The methodology seems to me to be very confusing and still needs to be restructured.
Lines 94 to 99 cause confusion to what you intend to investigate. It is important that the first paragraph of the Methodology describes the type of research that will be carried out.

Response

This has been revised thus:

A multi-stage approach was utilized, with Phase 1 focussing on the empirical findings on facilitators of, or barriers to prenatal physical participation in Buffalo City Municipality, Eastern Cape Province, which then lays the groundwork for the development of a prenatal physical activity strategy [24-27].

Comments

The objective presented at the end of the introduction indicates that physical activity intervention strategies will be developed and validated to encourage and promote prenatal physical activity, but in the first paragraph of the methodology it says that the development process and the validation of prenatal physical activity has been published elsewhere. So I believe that either the objective or the methodology is not adequate.

Response

What is alluded to in the methodology means the protocol outlining the development and validation of the physical activity strategy was published. This sentence has been corrected as per your recommendation in a later comment thus:

“A study protocol was previously published on the development and validation of physical activity intervention strategy for pregnant women in Buffalo City Municipality, South Africa [33].”

Comment

The sentence: “The empirical findings from both the quantitative and qualitative approaches, the 94 reported earlier, formed the foundation for the development of a prenatal physical activity 95 strategy (Phase I)” seems to me very confusing and does not reflect the type of study that is presented.

Response

This has been revised thus:

A multi-stage approach was utilized, with Phase 1 focussing on the empirical findings on facilitators of, or barriers to prenatal physical participation in Buffalo City Municipality, Eastern Cape Province, which then lays the groundwork for the development of a prenatal physical activity strategy [24-27].

Comment

The phrase “The process outling the development and validation of the prenatal physical activity has been published elsewhere [33].” must be rewritten. My suggestion: “A Study Protocol was previously published on the development of physical activity intervention strategy for pregnant women in Buffalo City County, South Africa.

Response

The sentence has been revised as suggested thus:

A study protocol was previously published on the development and validation of physical activity intervention strategy for pregnant women in Buffalo City Municipality, South Africa [33].

Comment

The study does not follow a standardized validation methodology. Kappa agreement analysis was not performed among the professionals evaluated. It should make clear which steps have already been taken and which have been carried out in this study.

Response

We earlier responded to this comment in your previous comment. We indicated that the study does not concern instrument development and validation; therefore, ‘kappa agreement analysis’ was not performed. The developed prenatal physical activity strategy was based, firstly, on empirical the findings on pregnant women’s levels, patterns, and associated factors related to prenatal activity, barriers, beliefs, knowledge, attitudes, perceived benefits and information sources, as well as the midwives’ perspectives on prenatal physical activity-related counselling. The next step was the development of the prenatal physical strategy by applying the SWOT, PESTLE and BOEM strategic models, which were explained in the methodology. Following the developed strategy, it was further validated with professional academic experts in prenatal physical activity and maternal health and key stakeholders, which include physicians, obstetricians, midwives, paediatric nurses, and primary healthcare nurses. The onus is on the policy managers/government to implement the developed and validated prenatal physical activity strategy in the setting to promote physical activity participation during pregnancy. 

Comment

Results and discussion
The results are unclear and the discussion does not reflect information from an instrument creation and validation study.

Response

The results of the stakeholders’ analyses of prenatal physical activity intervention strategy were presented in Table 2. We further presented their comments and the suggestions, which were incorporated into the strategy, accordingly. In addition, the stakeholders’ validation of the physical activity intervention strategies are presented in Table 3. Thereafter, by way of interpretation, we highlighted the strategies that were mostly endorsed by the health managers, midwives and pregnant women.

Regarding the discussion, it was focused on the key strategies commonly sanctioned by the stakeholders, with the relevant literature.

Reviewer 2 Report

The proposed changes highlighted in my previous review report have been applied and doubts have been clarified. Nice job. The manuscript can be accepted in the present form. 

Author Response

We thank you for accepting our revisions.

Reviewer 3 Report

This is an important manuscript focusing on the development and validation of a prenatal physical activity intervention strategy for South African women. The paper is well-written and presented in a structured way. There are a few points that I would like to rise in an attempt to further improve this manuscript. Please see below.

Given that the BOEM was added to the introduction, I suggest providing a few sentences to explain this model. It is only mentioned in the abstract and then in the aim (introduction), so it feels like the model was stated without preparing the reader for it.

Fig 1. Please consider changing the colour of the middle box (purple and red) to increase readability. I also suggest adding the full name of SWAT, PESTLE and BOEM in the figure legend, to make the figure as a standalone.

Please clarify: “In order to identify the relevant strengths and weaknesses, researchers conducted various interviews with experts, and consulted published and/or unpublished data.”

This statement needs to be further clarified. How many ‘experts’ were engaged in this study? Were they based on different locations? What level of seniority were they? It is also not clear what was the source of the unpublished data consulted.

In regards to the conclusion, restructuring medical and nursing curricula may be extremely challenging. These professionals already have heavy teaching/learning programs. I wonder if the authors would have any thoughts on offering professional development courses for health professionals involved in maternity care? Perhaps some courses could be partially or fully subsidised by the government?

Author Response

Reviewer #3: Comments and Suggestions for Authors

This is an important manuscript focusing on the development and validation of a prenatal physical activity intervention strategy for South African women. The paper is well-written and presented in a structured way. There are a few points that I would like to rise in an attempt to further improve this manuscript. Please see below.

Comment

Given that the BOEM was added to the introduction, I suggest providing a few sentences to explain this model. It is only mentioned in the abstract and then in the aim (introduction), so it feels like the model was stated without preparing the reader for it.

Response

The BOEM model has been explained thus:

The Build, Overcome, Explore and Minimize (BOEM) strategic model leverages on building strengths, overcoming weaknesses, exploring opportunities and minimizing threats, and in this case, factoring in the components of the BOEM to enhance prenatal physical uptake. The BOEM framework indicates the need to develop strategies that overcome mistrust, financial and human resources deficit, barriers to access and utilise physical activity facilities by pregnant women, while also eliminating stereotypical myths and beliefs, and minimising misuse of resources and ensure collaborative efforts by various stakeholders in promoting prenatal physical activity.

Comment

Fig 1. Please consider changing the colour of the middle box (purple and red) to increase readability. I also suggest adding the full name of SWAT, PESTLE and BOEM in the figure legend, to make the figure as a standalone.

Response

The colour of the middle box (purple and red) has been changed as suggested. The figure is a standalone and the full names of SWOT, PESTLE and BOEM were provided below the figure legend.

Comment

Please clarify: “In order to identify the relevant strengths and weaknesses, researchers conducted various interviews with experts, and consulted published and/or unpublished data.”

Response

In the preceding sentence, we explained the BOEM approach, which entails building on the identified strengths, overcoming the weakness, exploring the opportunities, and minimising the threats to physical activity. To accomplish these, interviews were conducted interviews with relevant experts on the subject, and also published or unpublished data were consulted in order to enable identification of the strengths and weaknesses of prenatal physical activity participation in the setting. In any case, the sentence has been revised to read: “In order to identify the relevant strengths and weaknesses of prenatal physical activity participation in the setting, the researchers conducted various interviews with experts, and consulted published and/or unpublished data from the Department of Health”.

Comment

This statement needs to be further clarified. How many ‘experts’ were engaged in this study? Were they based on different locations? What level of seniority were they? It is also not clear what was the source of the unpublished data consulted.

Response

We stated this in 2.1:  that “a purposive sample of seven professional academic experts with extensive knowledge, and proven academic and scholarly background on prenatal physical activity and maternal health, were selected to participate in the validation process”. They were in different universities in the Eastern Cape, which was the setting of the study.

Comment

In regards to the conclusion, restructuring medical and nursing curricula may be extremely challenging. These professionals already have heavy teaching/learning programs. I wonder if the authors would have any thoughts on offering professional development courses for health professionals involved in maternity care? Perhaps some courses could be partially or fully subsidised by the government?

Response

We appreciate this suggestion. The conclusion has been revised based on the suggestion accordingly.

“In addition, offering professional development courses for health professionals involved in maternity care in the Eastern Cape Province would empower healthcare providers with the necessary basic knowledge and requisite skills required to assist pregnant women with effective prenatal physical activity advice or conunselling”.

Reviewer 4 Report

Manuscript well-written, topic interesting and is an important one – urinary incontinence in women practicing exercise. The authors present the development and validation of the physical activity intervention strategy to encourage and promote prenatal physical activity and exercise. The manuscript was revised by the authors and resubmitted with alterations. I only have two minor considerations.

  1. The authors still have a long abstract (322 words).
  2. I'm not sure the authors understood reviewer 2's question (referring to the first comment in the results and discussion section), which I also have this concern. Gynecologists, obstetricians, midwives, and nurses can ONLY provide physical activity COUNSELING. I understand that most of the doctors were not familiar with the ACOG guidelines. Maybe, it could be clearer if you used another word like knowledge instead of training.

Author Response

Reviewer #4: Comments and Suggestions for Authors

Manuscript well-written, topic interesting and is an important one – urinary incontinence in women practicing exercise. The authors present the development and validation of the physical activity intervention strategy to encourage and promote prenatal physical activity and exercise. The manuscript was revised by the authors and resubmitted with alterations. I only have two minor considerations.

  1. Comment

The authors still have a long abstract (322 words).

Response

The abstract has been revised to 258 words.

Women rarely participate in physical activity during pregnancy, despite scientific evidence emphasising its importance. This study sought to develop intervention strategy to promote prenatal physical activity in Buffalo City Municipality, Eastern Cape Province, South Africa. A multi-stage approach was utilised. The Strength, Weakness, Opportunity and Threat (SWOT) approach was applied to the interfaced empirical findings on prenatal physical activity in the setting. Subsequently, the Build, Overcome, Explore and Minimise model was then used to develop strategies based on the SWOT findings. A checklist was administered to key stakeholders to validate the developed strategies. Key strategies to promote prenatal physical activity include: the application of the Mom-Connect (a technological device already in use in South Africa, to promote maternal health-related information for pregnant women) in collaboration with cellphone and network companies, the South African government to integrate prenatal physical activity and exercise training in the medical and health curricula to empower the healthcare providers with relevant knowledge and skills to support pregnant women in prenatal physical activity counselling, provision of increased workforce and the infrastructure necessary in antenatal sessions, and antenatal physical exercise classes and counselling, and the government, in partnership with various stakeholders, to provide periodical prenatal physical activity campaigns based in local, community town halls and clinics to address the lack of awareness, misrepresentations and concerns regarding the safety and benefits of physical activity during pregnancy. The effective implementation of this developed prenatal physical activity by policymakers and health professionals may help in the promotion of physical activity practices in the context of women in the setting.

  1. Comment

I'm not sure the authors understood reviewer 2's question (referring to the first comment in the results and discussion section), which I also have this concern. Gynecologists, obstetricians, midwives, and nurses can ONLY provide physical activity COUNSELING. I understand that most of the doctors were not familiar with the ACOG guidelines. Maybe, it could be clearer if you used another word like knowledge instead of training.

Response

This has been revised to read thus:

“Provide knowledge to gynaecologists, obstetricians, midwives, and nurses on prenatal physical activity counselling during antenatal sessions.”

Round 2

Reviewer 1 Report

Thanks for your effots. However several issues were not addressed by the authors.

This manuscript is a resubmission of an earlier submission. The following is a list of the peer review reports and author responses from that submission.

Round 1

Reviewer 1 Report

Thank you for giving me the opportunity to review this manuscript. This manuscript, entitled "Development and validation of prenatal physical activity intervention strategy for women in Buffalo City Municipality, 3 South Africa", aimed to present the development and validation of the physical activity intervention strategy to encourage and promote prenatal physical activity and exercise in the Eastern Cape context.

Summary:
It is not clear what the research objective, the methods, the results presented and the conclusions are. The results seem to me more like a description in the methodology.
The objectives of the abstract are not the same as those presented at the end of the introduction.
“The South African government to integrate prenatal physical activity and exercise training in the medical and health curriculum to empower the healthcare providers with relevant knowledge and skills to support pregnant women in terms of prenatal physical activity prescription and counseling.”. That phrase seems to be a recommendation.
The conclusion does not reflect the results of the studies. The authors highlight the need for future studies however this should not reflect research conclusions.
The summary needs to be reformulated.
Introduction. Lines 53-55. The authors report "that researchers have conducted multiple studies across countries and regions." However, no reference was cited.
Methods
I have doubts about the methods and results presented.
This study is an instrument development and validation study. However, it does not follow a standardized validation methodology. Kappa agreement analysis was not performed among the professionals evaluated. It should make clear which steps have already been taken and which steps have been taken in this study.

Reviewer 2 Report

This is an interesting article focusing on the development and validation of a prenatal physical activity intervention strategy for women in Buffalo city municipality in South Africa. In general, the manuscript is well written and clearly structured. However, I would like to draw the attention of the authors to some points and methodological issues in an attempt to improve this manuscript. Please find specific comments below.

Abstract

Based on the instructions for authors provided by Healthcare Journal, the abstract should be a single paragraph of about 200 words maximum and structured without headings. Please revise the abstract according to these instructions.

Abstract and main text: If I understand correctly, the first phase of the study is based on findings from previous publications of the authors (i.e references 16-19). This is not clear to the reader, especially in the abstract. For example, you mention that “A mixed methods design in a concurrent triangulation of both quantitative and qualitative approaches was utilized”. Did you use this methodology in the present article or in your previously published work?  

The results are summarized as if “bullet points” are used in a PowerPoint presentation. Please rewrite the main findings in a more formal way.

Methods section

If I am correct, the methods used in your study are based on your previous work “Okafor, U.B.; Goon, D.T. Developing a Physical Activity Intervention Strategy for Pregnant Women in Buffalo City Municipality, South Africa: A Study Protocol. Int. J. Environ. Res. Public Health 202017, 6694. https://doi.org/10.3390/ijerph17186694”.  This paper should be cited in the present manuscript. Moreover, you should describe in brief the methodology used in every phase in the present paper, especially regarding the data analysis employed. Please explain in the data analysis section how the comments from the experts were analyzed, which strategies that proved irrelevant were removed, while others were merged etc.

Results and discussion section

Table 2: Based on the first experts’ comment/critique the healthcare professionals such as gynecologists, obstetricians, midwives, and nurses can only provide physical activity counseling. Why did you include in the developed physical activity intervention strategy (Intervention strategy 1: Building on strength, Table 3 and in the discussion section) that these healthcare professionals should be trained to provide physical activity during antenatal sessions?  I agree with the experts’ comment that this is beyond their scope of practice. In most countries, physical activity is provided by exercise professionals and/or clinical exercise physiologists.  

Discussion section

Even though the discussion is clear and deep, a paragraph could be included in the discussion section comparing your findings with similar studies performed in other countries that guide interventions and the promotion of physical activity among pregnant women. Moreover, you can discuss whether the findings of your study can be generalized and applied in different healthcare systems around the world.

Minor comments:

Line 48: “…which are widely reported in the literature ([1-7].” Please delete the parenthesis.

Line 53-54: “Consequently, researchers have conducted various studies across countries and regions…” Please add some references.

Lines 59-61: As mentioned in a previous comment above, it should be clear to the reader that the empirical findings are not addressed in this particular work and have been reported in your previous publications (i.e. references 16-19).

Line 69: “therefore, a strategy to promote physical….” Add activity following physical.

Lines 92-93: “within the context of the Eastern Cape by applying the SWOT and PESTLE strategy models as part of the develop…” I propose that the BOEM model could be added to this sentence.

Figure 1 (Yellow Box) “Analysis of findings and identifying internal and external factors that could be…” A word is missing at the end of the phrase.  

Table 1 Title: “STRENGHTS AND WEAKNESS” should be “STRENGTHS AND WEAKNESSES”.

Line 146 “Validation is as technique,…”. Requires editing of English language.

Table 2 “Intervention strategy 1: Building on strength” is missing in Table 2.

Table 2 “Biokeneticists”. Do you mean biomechanists? Please replace it throughout the manuscript.

Table 2: Please add the word “on” before “their airtime slips” (Intervention strategy 3: Exploring opportunities).

Table 3 is not cited in the main text.